# Exploring the prokaryote-eukaryote interplay in microbial mats from an Andean athalassohaline wetland

Carolina F. Cubillos,[1] Pablo Aguilar,[2,3,4] David Moreira,[1] Paola Bertolino,[1] Miguel Iniesto,[1] Cristina Dorador,[2,3] Purificación López-García[1]

**ABSTRACT** Microbial community assembly results from the interaction between biotic and abiotic factors. However, environmental selection is thought to predominantly shape communities in extreme ecosystems. Salar de Huasco, situated in the high-altitude Andean Altiplano, represents a poly-extreme ecosystem displaying spatial gradients of physicochemical conditions. To disentangle the influence of abiotic and biotic factors, we studied prokaryotic and eukaryotic communities from microbial mats and underlying sediments across contrasting areas of this athalassohaline ecosystem. The prokaryotic communities were primarily composed of bacteria, notably including a significant proportion of photosynthetic organisms like Cyanobacteria and anoxygenic photosynthetic members of Alpha- and Gammaproteobacteria and Chloroflexi. Additionally, Bacteroidetes, Verrucomicrobia, and Deltaproteobacteria were abundantly represented. Among eukaryotes, photosynthetic organisms (Ochrophyta and Archaeplastida) were predominant, alongside relatively abundant ciliates, cercozoans, and flagellated fungi. Salinity emerged as a key driver for the assembly of prokaryotic communities. Collectively, abiotic factors influenced both prokaryotic and eukaryotic communities, particularly those of algae. However, prokaryotic communities strongly correlated with photosynthetic eukaryotes, suggesting a pivotal role of biotic interactions in shaping these communities. Co-occurrence networks suggested potential interactions between different organisms, such as diatoms with specific photosynthetic and heterotrophic bacteria or with protist predators, indicating influences beyond environmental selection. While some associations may be explained by environmental preferences, the robust biotic correlations, alongside insights from other ecosystems and experimental studies, suggest that symbiotic and trophic interactions significantly shape microbial mat and sediment microbial communities in this athalassohaline ecosystem.

**IMPORTANCE** How biotic and abiotic factors influence microbial community assembly is still poorly defined. Here, we explore their influence on prokaryotic and eukaryotic community assembly within microbial mats and sediments of an Andean high-altitude polyextreme wetland system. We show that, in addition to abiotic elements, mutual interactions exist between prokaryotic and eukaryotic communities. Notably, photosynthetic eukaryotes exhibit a strong correlation with prokaryotic communities, specifically diatoms with certain bacteria and other protists. Our findings underscore the significance of biotic interactions in community assembly and emphasize the necessity of considering the complete microbial community.

**KEYWORDS** community assembly, microbial mat, extreme environment, bacteria, protist, algae, biotic driver, co-occurrence network

Address correspondence to Purificación López-García, puri.lopez@universite-paris-saclay.fr.

Carolina F. Cubillos and Pablo Aguilar contributed equally to this article. The author order was determined by their contribution to the article.

The authors declare no conflict of interest.

See the funding table on p. 14.

Microbial mats are complex microbial ecosystems that grow on the surface of soils, rocks, or sediments of diverse aquatic habitats (1, 2). These stratified

ecosystems accommodate diverse micro-habitats and ecological niches, as well as billions of microorganisms that exhibit wide morphological, phylogenetic, and metabolic diversity (2, 3). Microbial mat metabolism integrates autotrophic (notably photosynthetic) and heterotrophic activities related to the progressive degradation of organic matter. These, in turn, strongly depend on steep redox vertical gradients which, as a function of the local hydrochemistry, can sometimes promote mineral precipitation leading to microbialite formation (4–6). Typical metabolic guilds, from surface to depth, broadly include oxygenic photoautotrophs, aerobic heterotrophs, anoxygenic photoautotrophs, sulfide oxidizers, sulfate reducers, and methanogens, among others (1, 7, 8). These microbial ecosystems are usually dominated by prokaryotes, which are typically characterized by 16S rRNA gene amplicon sequencing (7, 9). However, the importance of eukaryotes and viruses in microbial mats is being increasingly recognized (e.g., references 5, 10–14).

Modern microbial mats are often considered analogs of the oldest recorded ecosystems on Earth (1, 15, 16), such that highly conserved energy and carbon pathways within their anoxic layers can inform about prevailing core metabolism in Precambrian ecosystems (17). Although microbial mats are less widespread now than they were at that time (18), they typically occur worldwide in diverse ecosystems with more or less extreme conditions in terms of salinity, temperature, or UV radiation, among others (3, 8, 19). The Andean Plateau (or "Altiplano"), in particular, is home to several closed basins harboring wetlands and lagoons, locally called "salares," which are a reservoir of microbial mats thriving under high UV-radiation and exhibiting saline to hypersaline conditions, often combined with high metal content, and extreme daily temperature variation (19, 20). Salar de Huasco, recently declared Natural Park (March 2023), is an athalassohaline ecosystem located at 3,800 m above sea level with high physicochemical spatial variability, notably in terms of salinity. Its aquatic systems range from freshwater to salt-saturated brines, resulting in different microbial habitats as permanent or ephemeral ponds (21, 22). Thus, Salar de Huasco provides an excellent opportunity to study, within close geographic proximity (same basin), how complex microbial communities assemble and vary as a function of local environmental parameters. Several studies have previously focused on the bacteria and archaea thriving in water and/or sediment of different areas in the *salar* (23–26) and, more recently, their viruses (27, 28). In addition to these surveys across space, some temporal studies point out to seasonal variation (29, 30). Some specific members of these microbial communities have been studied in more detail, such as *Exiguobacterium* sp., which is a halotolerant and arsenic-resistant species isolated from sediment (31–33), or the aerobic anoxygenic phototroph *Rhodobacter* sp. Rb3, isolated from microbial mats (34). However, thorough studies of the microbial mat communities across the *salar* are lacking, as well as information about the microbial eukaryotic component other than fungal spores (35).

In this study, we aimed at identifying the environmental drivers influencing the assembly of both, prokaryotic and eukaryotic communities from microbial mats and underlying sediment of various sites in the Salar de Huasco with different physicochemical conditions. Our analyses show that both, abiotic and biotic factors, determine microbial community assembly. Specifically, the interplay between prokaryotes and photosynthetic eukaryotes appears remarkably strong.

## MATERIALS AND METHODS

### Study area, sampling, and measurements of physicochemical parameters

Salar de Huasco (20°18′18″ S, 68°50′22″ W) includes peatlands (locally called "bofedales"), permanent and non-permanent lakes, hypersaline lagoons, and salt crusts (20, 36). Various sampling sites have been historically described within this basin (21), representing gradients from freshwater (e.g., site H0) to hypersaline (e.g., sites H4 and H6) ecosystems. In December 2019 (i.e., during the wet season), we collected microbial mat and sediment samples from four sampling zones (H0, H3, H4, and H6) located

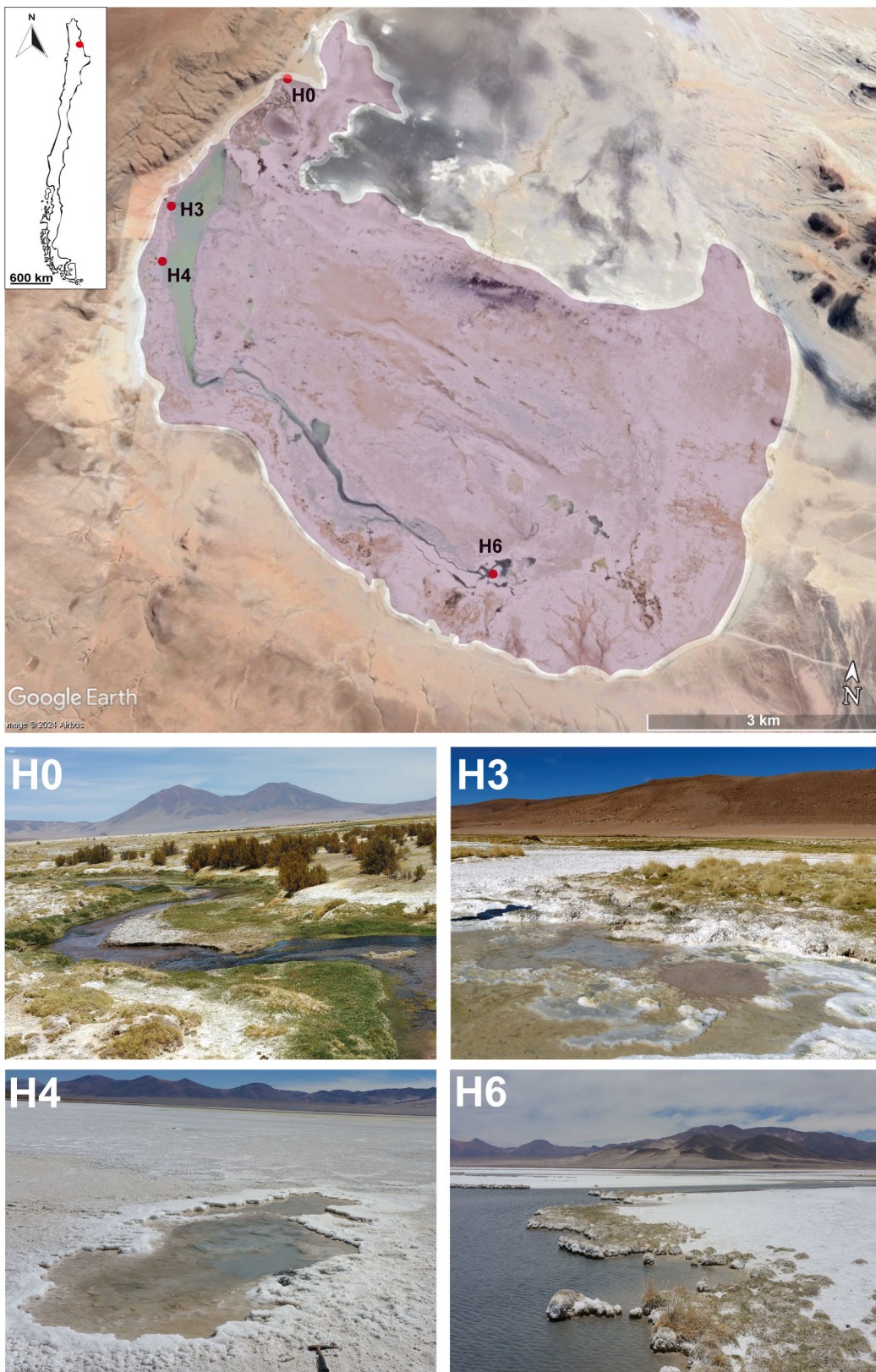

**FIG 1** Map of the study area indicating the sampling zones. Pictures of the sampling zones H0 to H6 are shown.

in Salar de Huasco (Fig. 1; Table S1). In this study, we denote as microbial mats the upper, layered microbial communities, usually cm-sized (1–5 cm), and as sediments, the undifferentiated, blackish, gray, or sandy material underneath the collected microbial mats (5–10 cm depth below the mat). In addition, a biofilm and a water sample were collected from sites H3 and H6, respectively. The biofilm, a much thinner (mm-sized) and apparently unstructured microbial community at macroscale as compared with the thick denser microbial mats, was floating on a pond. In total, we studied 19 samples from 11 sampling points distributed in the four sampling zones (Tables S1 and S2). Mat and biofilm samples were immediately fixed in absolute ethanol in Falcon tubes. About 5 L of water were passed through a 200-µm-pore-diameter nylon mesh, collected in a sterile bottle, and filtered in a nearby facility through a 0.2-µm-pore-diameter isopore polycarbonate filter (Merck Millipore, Darmstadt, Germany). The filter was then fixed in ethanol in a cryotube. Upon arrival at the French laboratory, samples were stored at −20°C until use. Physicochemical parameters were measured using a multiparameter probe Hanna HI9828 (Table S1).

## DNA extraction, PCR amplification, and sequencing

Genomic DNA was extracted using the PowerSoil DNA Isolation Kit (MoBio) after the elimination of ethanol and rehydration of the biomass at 4°C for at least 2 h in the resuspension buffer of the kit. DNA was used as a template for the V4-V5 region amplification (~420 bp fragments) of the prokaryotic 16S rRNA with the primer set U515F (5′-GTGCCAGCMGCCGCGGTAA-3′) and 926R (5′-CCGYCAATTYMTTTRAGTTT-3′). The eukaryotic 18S rRNA gene V4 region (~527 bp fragments) was amplified using the primers EK-565F (5′-GCAGTTAAAAAGCTCGTAGT-3′) and 18S-EUK-1134-R_UNonMet (5′-TTTAAGTTTCAGCCTTGCG-3′) biased against metazoans (37). Forward and reverse primers were tagged with 10 bp molecular identifiers (MIDs) to allow pooling and identification of amplicons from different samples. PCR-amplification reactions were performed in 25 µL volume reaction and contained 0.5–3 µL of eluted DNA, 1.5 mM of MgCl$_2$, 0.2 mM of deoxynucleotide (dNTP) mix, 0.3 µM of each primer, and 0.5 U of the hot-start Platinum Taq DNA Polymerase (Invitrogen, Carlsbad, CA). PCR reactions were carried out for 35 cycles (94°C for 30 s, 55–58°C for 30–45 s, and 72°C for 90 s) preceded by 2 min denaturation at 94°C and followed by five additional minutes of polymerization at 72°C. To minimize PCR bias, five different PCR reactions were pooled for each sample. Amplicons were then purified using the QIAquick PCR purification kit (Qiagen, Hilden, Germany). Sequencing was done using Illumina Miseq (2 × 300 bp, paired-end) at Eurofins Genomics (Konstanz, Germany).

## Amplicon data processing

A total of 1,585,086 raw sequences (reads) for 16S and 18S rRNA gene amplicons were obtained from 19 samples. The raw reads were analyzed using QIIME2 v.2020.8 (38). Briefly, the reads were demultiplexed using the "cutadapt demux-paired" function, and then, the DADA2 plugin was used to determine amplicon sequence variants (ASVs) after denoising, dereplication, and chimera-filtering (39). The optimal truncation parameters were selected using the program FIGARO (40). The forward and reverse reads were merged to obtain complete denoised ASV sequences. Denoised sequences with one or more mismatches in the overlap region were removed. Sequence and ASV statistics are provided in Table S2. ASVs were assigned to archaeal and bacterial taxa, based on the GTDB taxonomy (https://gtdb.ecogenomic.org/) release 207, using an in-house qiime classifier, which was trained and tested following qiime documentation (function feature-classifier) for prokaryotes; the curated database PR2 was used for eukaryotes (41). After the quality-filtering process, a total of 9,135 prokaryotic ASVs were obtained, 565 of which were archaeal ASVs. A total of 1,715 eukaryotic ASVs were determined across all samples. Sequences affiliating to chloroplasts, mitochondria, and Metazoa were removed from the analysis. The relative abundance and taxonomic affiliation of ASVs are provided in Table S3 and Table S4, respectively.

## Diversity, statistical, and network analyses

The estimates of species richness, Simpson, Shannon-wiener, and Pielou's diversity indexes were calculated using the "diversity alpha" function implemented in QIIME2 v.2020.8 (Table S2). Bray-Curtis distance (based on the relative abundance [percentage] of ASVs), ordinations (non-metric multidimensional scaling [NMDS] using the function metaMDS), the fit of environmental vectors into ordinations, and statistical differences among samples (ANOSIM) were calculated using the Vegan package in R (42). To the test whether the prokaryotic community was significantly correlated to the environmental variables and the eukaryotic community, we carried out Mantel tests using Spearman's correlation on the Bray-Curtis distance matrix and a matrix of Euclidean distances of the measured physicochemical parameters. Specifically, the physicochemical parameters included were pH, conductivity, total dissolved solids (TDC), and salinity. We applied a $\log(x + 1)$ transformation to these variables for data normalization and to minimize the impact of outliers. The transformed data were then systematically incorporated into a data frame, with each row corresponding to a different sample. This was used to construct an Euclidean distance matrix, reflecting the dissimilarities in environmental conditions across samples. This method effectively captures the multidimensional aspects of our environmental data. The Mantel test, executed with 9,999 permutations, compared this Euclidean distance matrix against the Bray-Curtis distance matrix of the prokaryotic community. To identify the core microbiome (i.e., microbial taxa shared by microbial mats and their underlying sediments), we used the package microbiome (DOI:10.18129/B9.bioc.microbiome) in R (42) applying an abundance-occurrence method with a threshold detection of 0.1% of relative abundance and persistence of 50%. We carried out a co-occurrence network analysis including prokaryotic and eukaryotic ASVs from mat and underlying sediment samples. The network was constructed using SpiecEasi with the Meinshausen and Bühlmann (mb) method (43). This method performs edge prediction according to the edge stability, inferred by the StARS model selection step (44). The set of high-confidence interactions (i.e., the top-ranked entries in the edge list; edge stability values > 0) were chosen according to the following criteria: negative edges ($n = 464$) were removed to visualize only co-occurrences, and edges were further filtered to exclusively show prokaryote-eukaryote co-occurrences. Additionally, we retained only the top 30% positively correlated edges (edge stability range 0–0.94). We used the fast-greedy clustering algorithm to detect clusters within the networks (45) and removed clusters with less than four nodes. In addition, the network topology measurements were calculated, including node and edge number, number of clusters, diameter, and modularity (Table S5). The network was visualized with the igraph R package (46).

## RESULTS AND DISCUSSION

### Alpha diversity of prokaryotic and eukaryotic communities

We studied the diversity of prokaryotes and microbial eukaryotes in microbial mats and underlying sediment (17 samples) from four major zones classically studied in the Salar de Huasco; two additional samples corresponded to plankton and one biofilm. These zones (H0 to H4) exhibit different physicochemical parameters (23, 24). Notably, at the sampling time, pH varied between 7 and 9, and salinity between 0.2 and 18.9 PSU (Table S1). The low salinity was associated with water inflow points in the *salar*. We observed a general salinity increase trend from the H0 zone, which is the main freshwater entry point, to the H4 zone, consistent with previous studies (24, 27). However, we also observed a high spatial heterogeneity, as has also been previously noted in Salar de Huasco (24, 29) and other high-altitude salt flats (47, 48). Thus, different points in the H4 zone exhibited different salinity values and, most notably, different sampling points had radically different levels of dissolved oxygen (Table S1). This variation from complete anoxia to highly oxic waters was related to the size and level of confinement of the different ponds, with smaller, deep ponds, being sometimes oxygen deprived.

We massively sequenced 16S and 18S rRNA gene amplicons to characterize, respectively, the prokaryotic (archaeal + bacterial) and eukaryotic diversity associated with these samples. We generated ASVs that were used to derive alpha diversity metrics (Table S2). The highest diversity values corresponded, as expected, to the prokaryotic component, with bacteria being prevalent. Thus, from the total 9,135 prokaryotic ASVs identified, 565 were archaeal. Only a small subset of those ASVs (2,864 bacterial, 38 archaeal) were shared between samples, which attests to the high heterogeneity of the different samples. From the total 1,715 eukaryotic ASVs identified, 570 were shared across samples; this higher percentage of shared ASVs was partly due to the occurrence of motile predatory protists (see below). Alpha diversity metrics for prokaryotes were not significantly different across samples (Shapiro-Wilk test, $P > 0.05$). However, the Simpson and Shannon-Wiener diversity indices, as well as Pielou's index (evenness) were significantly different (Shapiro-Wilk test, $P < 0.05$) between sample types, with higher values in the sediments as compared to the layered microbial mats on top of those sediments. The alpha diversity values for the eukaryotic microbial community were also higher for sediments, although the difference with other sample types was not significant (Shapiro-Wilk test, $P > 0.05$). We observed similar trends for prokaryotes and microbial eukaryotes when comparing among sampling sites.

## Prokaryotic and eukaryotic components of microbial communities

To get a phylogenetic insight into the microbial community structure of microbial mats and the underneath sediment, we attributed the ASVs to different taxa down to the genus level (whenever possible), and determined relative abundances (Tables S3 and S4). One biofilm floating in one of the ponds (H3.1.BF) and one plankton sample (H6.3.W.CT) were included for comparison (Table S2; Fig. 2). Bacteria largely dominated prokaryotic communities across samples (mean relative abundance of 96.2%), archaea representing the remaining, minor fraction of the community (Fig. 2A).

Overall, the most abundant bacterial high-rank taxa were the Bacteroidota (up to 41.8% of relative abundance), Alphaproteobacteria (up to 24.4%), and Gammaproteobacteria (up to 13.8%). However, their mean relative abundance in mat and sediment samples differed, as expected. Bacteroidota (29.8%), Alphaproteobacteria (15.1%), Gammaproteobacteria (11.1%), Cyanobacteria (8.3%), and Verrucomicrobia (4.1%) were more abundant in the microbial mat samples. The composition and relative abundance of archaeal and bacterial phyla dominant in Salar de Huasco at the order level are shown in Fig. S1. This distribution is consistent with the major metabolic functions associated with the two compartments. In the Salar de Huasco microbial mats, the photosynthetic cyanobacteria but also the anoxygenic photosynthetic members of the Alpha- and Gammaproteobacteria (essentially Rhodobacterales and Chromatiales, respectively) and, to a lesser extent, members of the Chloroflexales (Table S3; Fig. S1), contribute to the primary production. Among the identified genera of the Rhodobacterales (several ASVs likely define new genera), most corresponded to phototrophs (e.g., *Rubrimonas*, *Roseinatronobacter*, and *Roseovarius*), while some others include organisms metabolizing various organic compounds, such as *Tropicibacter*, grouping carbohydrate-utilizing, nitrate-reducing bacteria (49), *Yoonia*, grouping carbohydrate-utilizing organisms often associated to benthic algae (50) or the methylamine-metabolizing *Gemmobacter* (51). Among the Chromatiales, most identified genera corresponded to photoautotrophs (e.g., *Chromatium*, *Thiocapsa*, *Halochromatium*, *Thiorhodovibrio*, or *Allochromatium*; Table S3). Even some members of the consistently present but relatively low abundant Gemmatimonadota might include photosynthetic representatives (52, 53). The Bacteroidota were highly diverse and integrated only typical heterotrophic members of major classes Bacteroidia (Bacteroidales, Flavobacteriales, Chytinophagales, and Sphingobacteriales), Kapabacteria, Ignavibacteria, and Rhodothermia, among others (Fig. S1). They are most likely involved in the degradation of biopolymers produced by cyanobacteria and other photosynthetic members of the community. Like phototrophs, some of these Bacteroidota genera, such as *Rhodohalobacter*, relatively abundant in some Salar de Huasco

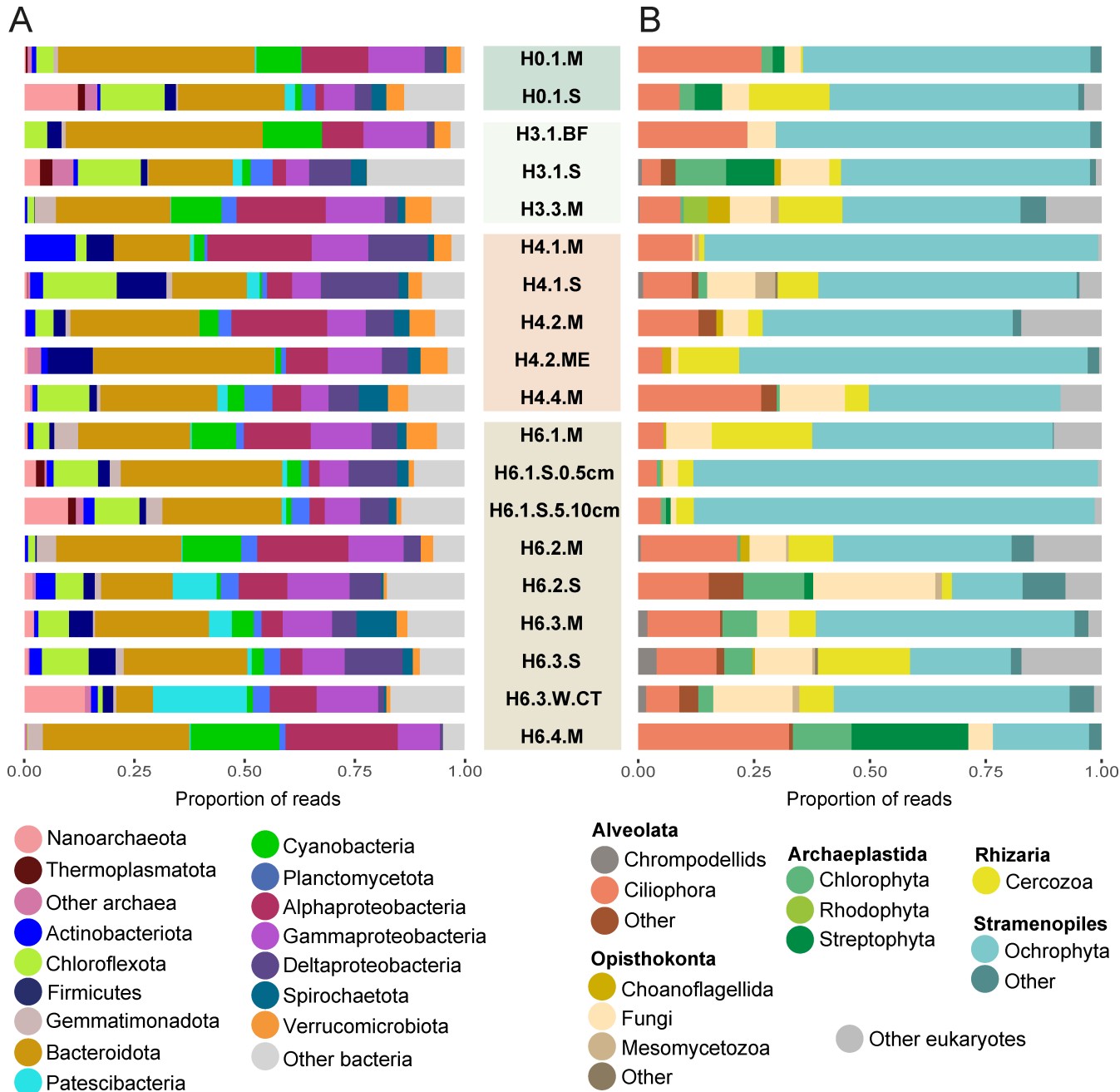

**FIG 2** Barplots showing the microbial community composition based of 16S/18S rRNA gene amplicon sequencing in samples from the Salar de Huasco. (A) Prokaryotic communities. (B) Eukaryotic communities. Samples labeled W, BF, M, and S correspond, respectively, to plankton, biofilm, microbial mat, and underlying sediment (Table S2).

samples (Table S3), are pigmented (54), which is adaptive in this highly irradiated environment. Bacteroidetes are well known for containing a wide diversity of enzymes to degrade glycans (55, 56), being typically associated with microbial mats (5, 47, 57, 58) and other carbohydrate-rich environments, including animal guts (59). Actually, some Chytinophagales positively correlate with green algal production (60) and some Ignavibacteria are known to be specifically cellulolytic (61). Some Kapabacteria seem to include potential sulfate reducers (62) and, accordingly, might potentially be responsible for the reduction of oxidized sulfur compounds produced by the sulfide-reducing

members of the alpha- and gammaproteobacterial photosynthesizers in the microbial mat compartment.

Although Bacteroidota and Alpha- and Gammaproteobacteria were also abundant in sediments, their proportion diminished as compared with the upper microbial mat layers, while heterotrophic members of the Chlorofexota (11.3%), Deltaproteobacteria (mostly Desulfobacterota, Bdellovibrionota, and Myxococcota; 9.4%), and Patescibacteria (2.8%) displayed higher relative abundances (Fig. 2A; Fig. S1). Patescibacteria were particularly abundant in the plankton sample (~20%). This bacterial taxon is highly diverse and comprises epibiotic, likely parasitic, bacteria (63). Frequent in subsurface waters, these bacteria seem more driven by host availability than environmental parameters (64, 65). Some Patescibacteria prey on photosynthetic gammaproteobacteria, such as *Vampirococcus lugosii* (Absconditabacterales) (66). Given the relative abundance of these types of anoxygenic photosynthesizers, they might be a potential target for some members of that patescibacterial order (Table S3). Deltaproteobacterial taxa, including among other newly defined phyla the Desulfobacterota, Bdellovibrionota, and Myxococcota, which were particularly diverse in Salar de Huasco mats (Fig. S1C; Table S3), are typically associated with microbial mat layers and sediments. Many Desulfobacterota are typical sulfate reducers that participate in the degradation of organics and the sulfur cycle (e.g., in our samples, *Desulfotignum*, *Desulfobacula*, *Desulfobacter*, *Desulfobacterium*, *Desulfobulbus*, *Desulfopila*, *Desulforhopalus*, *Desulfococcus*, *Desulfomicrobium*, *Desulfonatronum*, *Desulfatitalea*, *Desulfosudis*, among many others; Table S3) (67), although others, are involved in hydrogen-mediated syntrophy (e.g., in our samples, Syntrophobacteriales, Syntrophorhabdales, or Syntrophales; Table S3) (68). Bdellovibrionota and Myxococcota (also represented by multiple genera; Table S3) are known as typical bacterial predators (69), although they might even occasionally contribute to photosynthetic carbon fixation (70).

The archaeal community was dominated by Nanoarchaeota and Thermoplasmatota, which reached the highest mean proportion in sediment samples (4.2% and 1%, respectively), although Nanoarchaeota reached values around up to 10–13% in the sediment samples H0.1.S, H6.1.S.5.10cm and the plankton sample (Fig. 2A). Pacearchaeales and Woesearchaeales were the most abundant groups in these three samples (Fig. S1). The rest of the archaea were diverse across samples, albeit less relatively abundant, and comprised Thermoplasmatota (essentially the groups DHVEG-1, EX4484-6, SG8-5, and UBA10834), Halobacteriota (Halobacteriales, Methanomicrobiales, Methanosarcinales, and Methanotrichales), Thermoproteota (B26-1), and Asgardarchaeota (Lokiarchaeles, Helarchaeales, and Thorarchaeales) (Fig. S1).

The eukaryotic community was largely dominated by Ochrophyta (up to 87% of the protist component) in almost all samples (Fig. 2B). This group of photosynthetic stramenopiles included pennate diatoms (Bacillariophyta), golden algae (Chrysophyceae), yellow-green algae (Xantophyceae), and one Dyctiochophyceae ASV (Table S4; Fig. S2). Among the Archaeplastida, green algae (Chlorophyta) and Streptophyta were rather abundant, with minor occurrences of red algae (Rhodophyta). Rare Cryptophyta and Haptophyta ASVs completed the photosynthetic cohort of microbial eukaryotes in the Salar de Huasco samples. These groups were not only detected in the microbial mats but also in the underlying sediments, indicating that, although they can no longer perform photosynthesis, their DNA is preserved during burial. Diatom DNA appears to be particularly well preserved in ancient samples and can be used to retrace historical events (71). Alternatively, some of them might be using mixotrophic strategies. Ciliates, fungi and Cercozoa were the three more represented clades of heterotrophic eukaryotes. Ciliates were highly diverse, although the most represented groups were the Oxytrichidae and Sessilida (Table S4). Ciliates are usually abundant grazers feeding on benthic samples and are typically identified on microbial mats and freshwater systems (5, 72). Fungi were dominated by flagellated lineages (73), many of which may be parasites of other opisthokonts. Cercozoans comprised diverse lineages of predators including vampyrellids (Fig. S2), some of which are typical predators of algal lineages

(74) and members of the clade 10–12, which are eukaryvorous flagellates widespread in freshwater systems (75).

We looked for shared ASVs potentially conforming to a core microbiome in microbial mats and sediments. The prokaryotic core microbiome across space and sample types was small: 11 ASVs for microbial mats and 30 ASVs for sediments (Fig. S3). Four ASVs corresponded to photosynthetic genera in microbial mats, one alphaproteobacterium (Rhodobacteraceae), two gammaproteobacteria (genus *Thioploca*), and one cyanobacterium belonging to the genus *Symploca*. The latter genus groups cyanobacteria producing highly cytotoxic compounds (76), which might perhaps facilitate its colonization across samples. The prokaryotic sediment core comprised members of the Bacteroidota, Chloroflexota, Cyanobacteria, Gammaproteobacteria, Alphaproteobacteria, Deltaproteobacteria, Firmicutes, Spirochaetota, and Gemmatimonadota (Fig. S3). The protist core microbiome was also very limited. In microbial mats, it involved two diatoms and two ciliate ASVs; in the sediments, 11 diatom ASVs and one fungal ASV (Fig. S3). Overall, this reduced core of shared ASVs illustrated sample heterogeneity and suggested strong local environmental selection.

## Environmental drivers of prokaryotic and eukaryotic community assembly

Both abiotic and biotic factors contribute to microbial community assembly (77). In extreme environments, environmental selection is thought to play a key role but the influence of biotic interactions has been rarely studied in this type of systems. In our study, we tried to disentangle the influence of abiotic versus biotic factors on prokaryotic and eukaryotic members of the Salar de Huasco microbial communities. To do so, we first looked for the effect of the measured physicochemical parameters on community structure. Ordination analyses (NMDS) of prokaryotic and eukaryotes communities showed relatively dispersed communities, even though samples from the same sampling zone tended to group together (Fig. 3). We applied ANOSIM tests to see whether microbial communities significantly differed among samples. This was the case for prokaryotic communities, which differed among sampling sites (ANOSIM, $R = 0.3$, $P = 0.01$) and sample types (ANOSIM, $R = 0.24$, $P = 0.02$). Furthermore, prokaryotic communities segregated according to the salinity gradient exhibited across sampling

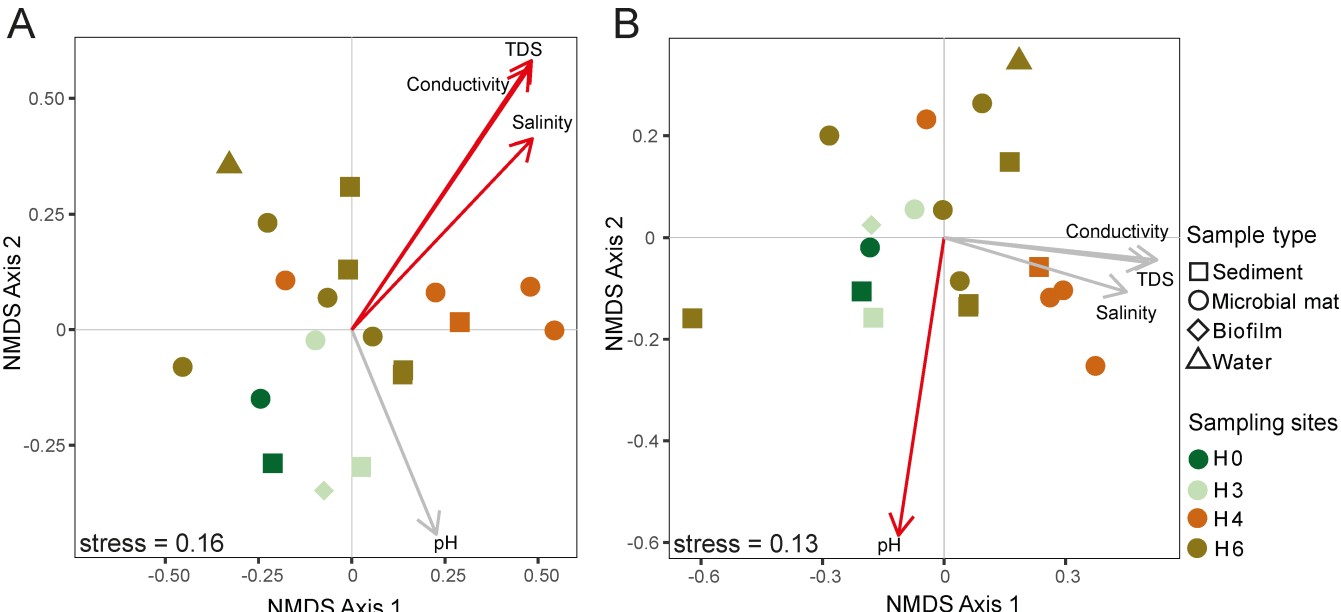

**FIG 3** Nonmetric multidimensional scaling analysis of microbial communities studied in the Salar de Huasco. (A) Prokaryotic communities. (B) Eukaryotic communities. Vectors of major measured environmental parameters (Table S1) are included; significant parameters are shown with red arrows. TDS, total dissolved solids.

sites (Fig. 3A). Salinity ($R^2 = 0.4$) and the related parameters, total dissolved solids (TDS; $R^2 = 0.57$) and electrical conductivity ($R^2 = 0.55$), had a significant effect on the segregation of the prokaryotic fraction. By contrast, global eukaryotic communities did not show significant differences across sampling sites ($R = 0.13$, $P = 0.15$) or sample types ($R = 0.07$, $P = 0.26$). However, pH appeared to explain, albeit with modest probability ($R^2 = 0.35$, $P = 0.041$), the distribution of eukaryotic communities on the ordination analysis (Fig. 3B).

We then looked for the cumulative effect of measured environmental factors (Table S1) on the community structure, measured via Bray-Curtis distances, both for prokaryotes and eukaryotes, using Mantel tests. We excluded dissolved oxygen, which was only measured in the water column above the mat and sediment samples. Taken together, these abiotic factors significantly correlated with prokaryotic and eukaryotic components of microbial communities, although the correlation was stronger for prokaryotes (Fig. 4A) than for eukaryotes (Fig. 4B). Since motile predatory protists, such as ciliates or cercozoans, which can move between sites and are not strictly associated to benthic communities, constituted a considerable fraction of the eukaryotic communities, we also carried out independent tests for photosynthetic and non-photosynthetic eukaryotes. While algal communities showed a significant, albeit weak, correlation with environmental parameters (Fig. 4C), heterotrophic eukaryotes, as expected, did not correlate with differences of abiotic parameters across Huasco sites (Fig. 4D).

Finally, we looked for a potential mutual influence of prokaryotic and eukaryotic communities. Mantel tests showed a marked significant correlation of prokaryotic communities and both, photosynthetic (Fig. 4E) and non-photosynthetic (Fig. 4F) eukaryotes. However, the correlation of prokaryotes with heterotrophic eukaryotes was somewhat weaker and apparently driven by some outlier points (Fig. 4F). By contrast, the correlation of prokaryotes and benthic algae was much stronger and even (Fig. 4E), suggesting a strong interplay between these two components of microbial communities across Salar de Huasco samples.

## Co-occurring patterns in mat and sediment communities

To identify more specific potential interactions between prokaryotes and eukaryotes in microbial mats and sediments, we built co-occurrence networks. The global co-occurrence network was complex (Fig. S4) and, to facilitate the visualization of inter-domain co-occurring ASVs, we filtered out negative correlations and retained only the top 30% positive correlations that contained both eukaryotes and prokaryotes (Fig. 5). This network consisted of 116 nodes (69 bacterial and 47 eukaryotic; Tables S5 and S6). Without surprise, archaeal nodes were not identified under the chosen filters. Using a fast greedy algorithm, we identified 19 clusters within the co-occurring network. Eukaryotes, essentially Ochrophyta, displayed the highest number of edges in nine clusters, whereas bacteria, notably Proteobacteria, were central in only three clusters (Fig. 5; Table S6).

While some co-occurrences might be due to similar preferences for specific environmental conditions, some might actually reflect trophic or symbiotic (mutualistic, commensal, or parasitic) interactions. In many cases, disentangling the two possibilities (environmental preference vs biotic interactions) will require experimental evidence; yet, identifying these potential interactions may help establishing working hypotheses and orient further studies. The stronger correlation between prokaryotes and photosynthetic eukaryotes (Fig. 4E) indeed suggested a role for biotic interactions. In Salar de Huasco, diatoms were the most represented ochrophytes and they co-occurred with cyanobacteria and their typically associated glycan-degrading Bacteroidota in several clusters, notably clusters 3, 5, and 11. Diatoms might co-occur with cyanobacteria because they need light and similar specific conditions. However, symbioses between diatoms and $N_2$-fixing cyanobacteria are widespread in marine and other ecosystems (78, 79), suggesting that this type of mutualism might be operational in Huasco as well. Some of the cyanobacterial ASVs co-occurring with diatoms were not assigned to described genera and it is difficult to predict whether they represent $N_2$-fixers, especially since $N_2$ fixation, although widespread, is patchily distributed among cyanobacterial taxa (80). However, some ASVs

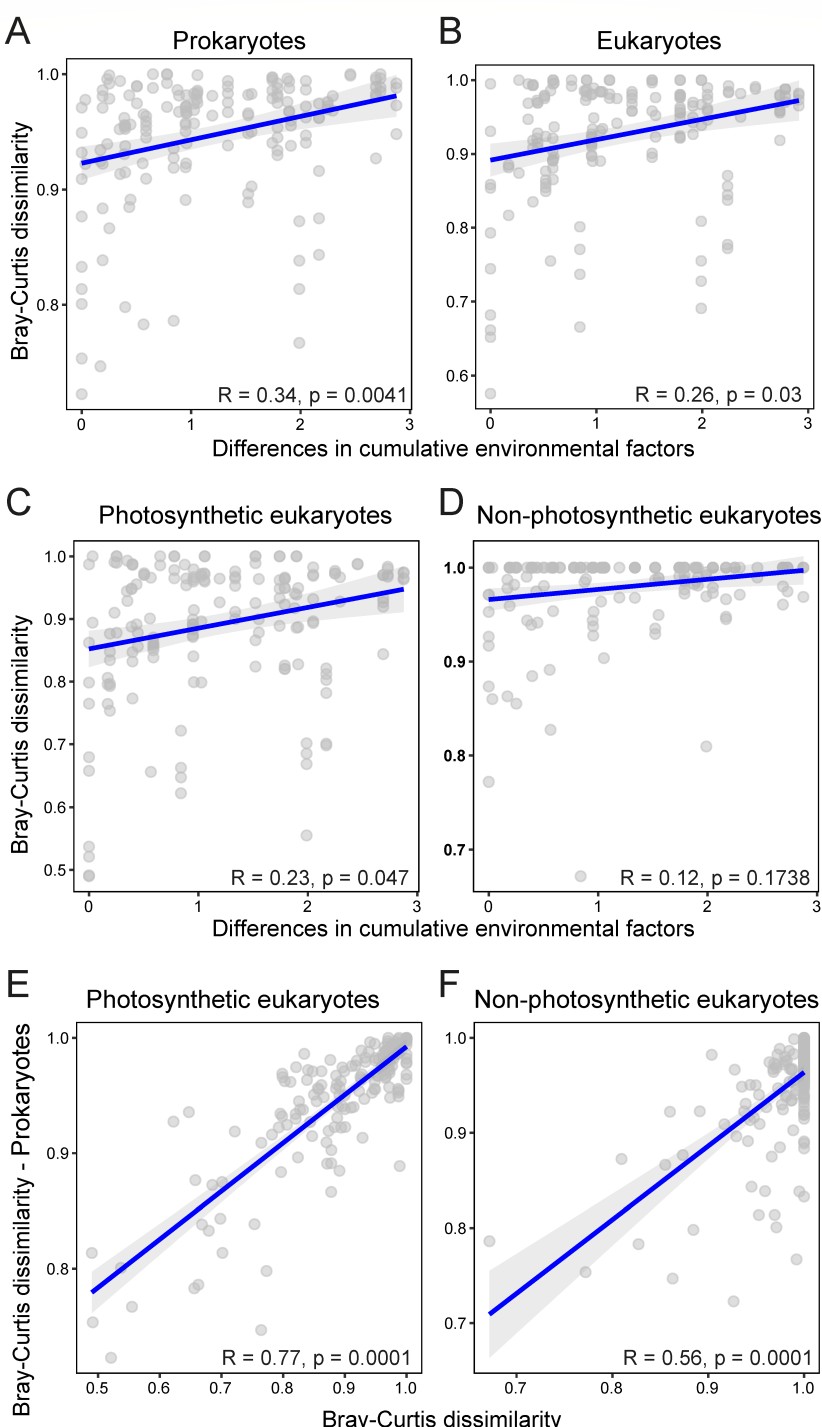

**FIG 4** Results of Mantel tests showing correlations of microbial communities with abiotic and biotic parameters. (A and B) Correlations between prokaryotic (A) and eukaryotic (B) community structure (Bray-Curtis dissimilarities) as a function of cumulative abiotic environmental parameters. (C and D) Correlations between the photosynthetic (C) and non-photosynthetic (D) component of eukaryotic communities as a function of cumulative abiotic environmental parameters. (E and F) Correlation between prokaryotic community structure and photosynthetic (E) and non-photosynthetic (F) fractions of eukaryotic microbial communities.

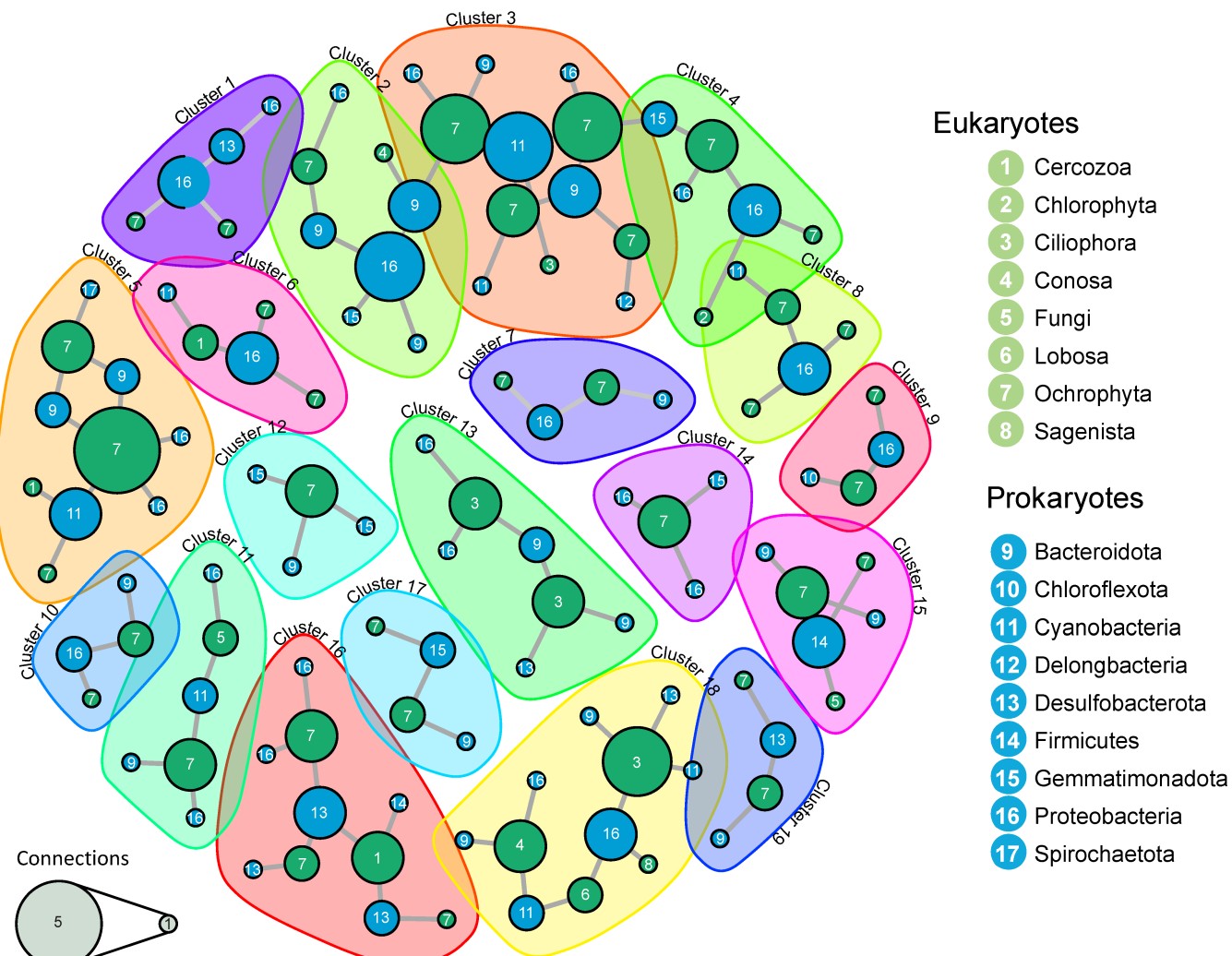

**FIG 5** Co-occurrence networks of prokaryotic and eukaryotic members of mat and sediment samples from Salar de Huasco. Only the top 30% of positive correlations between prokaryotic and eukaryotic ASVs are displayed.

could be assigned to known genera of $N_2$-fixers, such as *Dolichospermum,* which co-occurs with diatoms in cluster 11 (81) (Table S6; Fig. 5). Furthermore, $N_2$ fixation is not exclusive of cyanobacterial communities but can also be carried out by some anoxygenic phototrophs (e.g., several Proteobacteria and Chlorobi) in their respective (micro)oxic and anoxic microbial mat niches (82), therefore bearing a potential for metabolic exchange with diatoms. Thus, different clusters associated diatoms with anoxygenic photosynthetic proteobacteria, for example, clusters 1 and 10, the latter including *Thiocapsa*, a genus also bearing $N_2$ fixers (83). Gemmatimonadetes, which also comprise some phototrophic organisms (84), were also often associated with diatoms (clusters 4, 12, 14, or 17). Interestingly, these bacteria can attach to diatoms in the epilimnion of some lakes (85).

In addition to potential symbiotic interactions, some of the observed co-occurrences might reflect trophic interactions and prey preferences. For instance, ciliates and some amoeba (lobosa and conosa) were frequently associated with Proteobacteria and Bacteroidota or Cyanobacteria (clusters 13 and 18). Some other algal predatory protists, such as vampyrellids (86), occurred in clusters containing diatoms or other ochrophytes (clusters 5, 6, and 16). Desulfobacterota also appeared in several clusters (Fig. 5). In summary, at finer taxonomic levels, our study suggests that, beyond the threshold of

environmental selection, several specific trophic, and symbiotic interactions shape Salar de Huasco microbial mat and sediment communities.

## Concluding remarks

Both abiotic and biotic factors contribute to microbial community assembly (77). However, for microbial communities, environmental selection by abiotic parameters is often thought to be determinant since early times and the Baas-Becking's "everything is everywhere but the environment selects" tenet that imprinted the Delft's school of microbiology (87, 88). Thus, environmental selection is thought to be dominant especially for extreme environments such as deserts, hypersaline or hot settings, where abiotic selective pressures are strong (e.g., references 89–92). Nonetheless, on top of the potential role of stochasticity (93), it is becoming clear that functional selection imposed by other community members (94, 95) and other trophic interactions (96), which may take place at very small scale (97), are also key drivers of microbial community assembly (96). Here, we tried to disentangle the influence of abiotic versus biotic factors on prokaryotic and eukaryotic members of microbial mats and underlying sediments in the Salar de Huasco, which qualifies as a poly-extreme extreme environment and exhibits spatial physicochemical gradients (20–22). 16S/18S rRNA gene amplicon studies revealed widely diverse prokaryotic, and to a lesser extent, eukaryotic communities. The photosynthetic component of these communities was particularly abundant and diverse, with both oxygenic (cyanobacteria, and eukaryotic algae—essentially ochrophytes and Archaeplastida) and anoxygenic photosynthesizers (several Alpha- and Gammaproteobacteria and Chloroflexi). Despite the presence of similar high-rank taxa (Fig. 2), lower-rank taxa (ASVs) differed among sites, with very reduced core prokaryotic and eukaryotic microbiomes, suggesting strong local environmental selection. In particular, salinity influenced the most prokaryotic community assembly (Fig. 3). Globally, the cumulated effect of measured environmental parameters significantly correlated with prokaryotic and protist communities. However, within eukaryotes, we observed a marked difference between the photosynthetic and the heterotrophic fraction, the latter not displaying any significant correlation with the environmental factors (Fig. 4). Benthic algae, little or no motile, seemed to be shaped by the local physicochemistry and behave very much like the prokaryotic component of microbial mats and sediments. By contrast, heterotrophic protists, such as ciliates, amoeba, and cercozoans, are flexible grazers able to move between different areas in the Salar. Interestingly, the influence of biotic factors seemed to be much higher than that of abiotic parameters. In particular, prokaryotic communities strongly correlated with photosynthetic eukaryotes (Fig. 4). Co-occurrence networks suggest potential interactions between specific community members, for instance, between diatoms and specific photosynthetic and heterotrophic bacteria, or protist predators. While this influence could be partly explained by similar environmental preference, the strong biotic correlation together with known interactions from similar partners in other ecosystems (78, 79, 85, 86), suggest that symbiotic (from mutualism to parasitism) and trophic and interactions shape the microbial mat and sediment communities in this athalassohaline ecosystem. Our study highlights the importance of considering the entire microbial community, including eukaryotic microorganisms, to better understand community assembly. Further studies focusing on the functional role of specific microbial interactions should provide deeper insights into the microbial ecology of the Salar de Huasco and other similar athalassohaline systems. The knowledge gained from the Andean Altiplano *salares*, severely threatened by mining activities (98), can also aid in preserving these unique microbial ecosystems.

## ACKNOWLEDGMENTS

This work was supported by the bilateral French-Chilean ECOS Sud-ANID cooperation program (project MMEX), the ERC AdG PlastEvol (787904, D.M.) and the Moore-Simons Project on the Origin of the Eukaryotic Cell, Moore Foundation grant GBMF9739 (P.L.-G.;

[https://doi.org/10.37807/GBMF9739](https://doi.org/10.37807/GBMF9739)). P.A. received funding from ANID–Millennium Science Initiative Program–NCN2021-056.

## AUTHOR AFFILIATIONS

[1]Ecologie Systématique Evolution, CNRS, Université Paris-Saclay, Gif-sur-Yvette, France
[2]Laboratorio de Complejidad Microbiana, Instituto Antofagasta and Centro de Bioingeniería y Biotecnología (CeBiB), Universidad de Antofagasta, Antofagasta, Chile
[3]Departamento de Biotecnología, Facultad de Ciencias del Mar y Recursos Biológicos, Universidad de Antofagasta, Antofagasta, Chile
[4]Millennium Nucleus of Austral Invasive Salmonids - INVASAL, Concepción, Chile

## AUTHOR ORCIDs

Purificación López-García ⓘ http://orcid.org/0000-0002-0927-0651

## FUNDING

| Funder | Grant(s) | Author(s) |
| --- | --- | --- |
| Gordon and Betty Moore Foundation (GBMF) | GBMF9739 | Purificación López-García |
| EC \| European Research Council (ERC) | 787904 | David Moreira |

## AUTHOR CONTRIBUTIONS

Carolina F. Cubillos, Data curation, Formal analysis, Methodology, Writing – original draft | Pablo Aguilar, Data curation, Formal analysis, Methodology, Writing – original draft | David Moreira, Funding acquisition, Supervision | Paola Bertolino, Methodology | Miguel Iniesto, Data curation, Formal analysis | Cristina Dorador, Resources | Purificación López-García, Conceptualization, Funding acquisition, Investigation, Methodology, Supervision, Visualization, Writing – original draft, Writing – review and editing

## DATA AVAILABILITY

16S and 18S rRNA gene amplicon sequences have been deposited in the GenBank (NCBI) Sequence Read Archive with BioProject number PRJNA1046993.

## ADDITIONAL FILES

The following material is available online.

### Supplemental Material

**Supplemental material (Spectrum00072-24-s0001.pdf).** Tables S1, S2, S5, and S6 and supplemental figures.
**Supplemental tables (Spectrum00072-24-s0002.xlsx).** Tables S3 and S4.

### Open Peer Review

**PEER REVIEW HISTORY (review-history.pdf).** An accounting of the reviewer comments and feedback.

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
