## [Reviewer comments · Microbiology Spectrum]

Microbiology Spectrum

Exploring the prokaryote-eukaryote interplay in microbial mats from an Andean athalassohaline wetland

Purificación López-García, Carolina Cubillos, Pablo Aguilar, David Moreira, Paola Bertolino, Miguel Iniesto, and Cristina Dorador

Corresponding Author(s): Purificación López-García, CNRS Delegation Ile-de-France Sud

Review Timeline:

Submission Date:

January 10, 2024

Accepted:

January 28, 2024

Editor: Adriana Lopes dos Santos

Reviewer(s): The reviewers have opted to remain anonymous.

Transaction Report:

DOI: <https://doi.org/10.1128/spectrum.00072-24>

Re: Spectrum00072-24 (Exploring the prokaryote-eukaryote interplay in microbial mats from an Andean athalassohaline wetland)

Dear Dr. Purificación López-García:

Your manuscript has been accepted, and I am forwarding it to the ASM production staff for publication. Your paper will first be checked to make sure all elements meet the technical requirements. ASM staff will contact you if anything needs to be revised before copyediting and production can begin. Otherwise, you will be notified when your proofs are ready to be viewed.

Sincerely,
Adriana Lopes dos Santos
Editor
Microbiology Spectrum